# A Systematic Review and Meta-Analysis: *Lactobacillus acidophilus* for Treating Acute Gastroenteritis in Children

**DOI:** 10.3390/nu14030682

**Published:** 2022-02-06

**Authors:** Haixin Cheng, Yi Ma, Xiaohui Liu, Chao Tian, Xuli Zhong, Libo Zhao

**Affiliations:** 1Department of Pharmacy, Children’s Hospital of Capital Institute of Pediatrics, Beijing 100020, China; chhaixin@163.com; 2Department of Pharmacy, Peking University Third Hospital, Beijing 100191, China; mayi@bjmu.edu.cn; 3Department of Pharmacy, Beijing Children’s Hospital, Capital Medical University, Beijing 100045, China; liuxiaohui5077@163.com (X.L.); 18811213680@163.com (C.T.)

**Keywords:** *Lactobacillus acidophilus*, acute gastroenteritis, children, systematic review, meta-analysis

## Abstract

The efficacy of probiotic strains of *Lactobacillus acidophilus* to manage acute gastroenteritis in children is still not established. We searched the Cochrane Library, PubMed, EMBASE, and three Chinese literature databases (CNKI, WanFang, and CBM) from their inception to February 2021 for RCTs that compared the use of *Lactobacillus acidophilus* with no *Lactobacillus acidophilus*. The grey literature was searched through Google Scholar. Authors of the original papers were contacted for additional data. The study included a total of 15 RCTs involving 1765 patients. Compared with placebo or no treatment, *Lactobacillus acidophilus* was associated with a reduced duration of diarrhea (moderate quality of evidence), but the effect was not statistically significant when only the individual probiotic strain was provided. *Lactobacillus acidophilus* was effective when used at a daily dose ≥ 10^9^ CFU. There was no difference in the effect of *Lactobacillus acidophilus* on diarrhea duration among Asian, European, or American countries. *Lactobacillus acidophilus* reduced the frequency of diarrhea on day 2 to day 5. However, it was statistically significant on day 3. When administered at a dosage of more than 10^9^ CFU to children with acute gastroenteritis, moderate- to low-quality data showed that *Lactobacillus acidophilus* reduced the duration of diarrhea and conferred a benefit for frequency of diarrhea.

## 1. Introduction

For children under the age of five, acute gastroenteritis (a clinical illness characterized by increased stool frequency, accompanied with or without vomiting, fever, or stomach discomfort) ranks second on the list of main causes of mortality in the world [1]. Despite a 90 percent reduction in diarrhea-related mortality over the previous forty years, acute gastroenteritis continues to be a serious public health concern. For example, in 2018, approximately 500,000 children died from diarrhea [2]. The *European Evidence-Based Guidelines* for managing acute gastroenteritis in children concluded that heat-killed *Lactobacillus acidophilus* LB (low quality of evidence, weak recommendation) demonstrated some efficacy in reducing acute gastroenteritis-related symptoms in pediatric age groups [3]. *Lactobacillus acidophilus* achieved maximum survival rates and CFU/mL after adaptation at pH 4.2 and 5.0 [4]; in consequence, it can easily reach the small intestine (pH < 7), and is able to thrive in the small intestine rather than the colon (pH > 7). Possible mechanisms for the therapeutic effects of *Lactobacillus acidophilus* include amelioration of the impairment of electrolyte absorption [5], maintenance of immunological homeostasis [6], antibiotic activity/antiviral activity [7], and promotion of intestinal epithelial integrity [8]. In addition, in 2018, a network meta-analysis of randomized and quasi-randomized trials found high-quality evidence that combinations of *Lactobacillus acidophilus* and other probiotic strains could reduce the duration of diarrhea by approximately one day (MD −26.3 h, 95% CI −16.2 to −36.1) compared with standard care or placebo [9].

However, in the light of recent null studies, there is still controversy as to whether the efficacy of *Lactobacillus acidophilus* has been established. A relatively large, prospective randomized controlled trial (RCT) carried out in Vietnam, including 290 children aged 11.8 months to 21.5 months, compared the operation of *Lactobacillus acidophilus*, at a dose of 4 × 10^8^ CFU twice daily, to a matching placebo, found no differences in outcomes, including duration of diarrhea and stool frequency in the first three days after enrollment, between groups [10]. Moreover, in several additional randomized controlled trials (RCTs), no difference in diarrhea duration was reported across groups [11,12]. The purpose of this study was to assess the effectiveness of *Lactobacillus acidophilus* supplementation for treating children with acute gastroenteritis, and to give suggestions on probiotic treatments for acute gastroenteritis in children according to the assessment result.

## 2. Materials and Methods

### 2.1. Study Design

The study protocol was designed by the authors H.C., Y.M. and L.Z. This report complies with the recommendations of the Preferred Reporting Items for Systematic Reviews and Meta-Analyses (PRISMA: http://www.prisma-statement.org/ (accessed on 21 February 2021)) guidance. The systematic review and meta-analysis protocol were registered with the PROSPERO (National Institute for Health Research, NIHR. http://www.crd.york.ac.uk/prospero/ (accessed on 21 February 2021)) international prospective register of systematic reviews (CRD42021254066).

### 2.2. Inclusion and Exclusion Criteria

All relevant RCTs that compared use of *Lactobacillus acidophilus* (as a single ingredient or a multispecies synbiotic mixture, in all formulations, at any dose, regardless of the strain manufacturer), with no *Lactobacillus acidophilus* (defined as placebo or no treatment), were eligible for inclusion. Patients were children (<18 years old) with acute gastroenteritis. Only RCTs published in English and Chinese were considered. The primary outcome measure was the duration of diarrhea. The secondary outcome measure was the frequency of diarrhea at various time intervals. Letters to the editor, abstracts and scientific conference proceedings were all omitted.

### 2.3. Search Strategy

To acquire relevant evidence, the Cochrane Library, PubMed, EMBASE and three Chinese literature databases (CNKI, WanFang, and CBM) were searched from their inception to February 2021. The principal search terms and keywords used were as follows: diarrhea/diarrhoea, diarrh*, gastroenteritis, probiotic*, *Lactobacillus acidophilus*, and randomized controlled trial. For full details, see Appendix A. Additionally, the grey literature was searched through Google Scholar. Some clinical trials databases (https://www.ClinicalTrials.gov (accessed on 21 February 2021) and https://www.ClinicalTrialsRegister.eu (accessed on 22 February 2021)) were also searched for RCTs. The reference lists from identified studies and key review articles were also searched if the title, abstract, name of substance word, subject heading word, keyword heading word, protocol supplementary concept word or unique identifier contained all of “*Lactobacillus acidophilus*” and “acute gastroenteritis” and “children”. If further information was required, the authors of the original papers were contacted for additional data.

### 2.4. Data Extraction

Study inclusion was determined by two investigators working separately, with the senior investigator addressing any discrepancies between the two investigators. The same investigators independently extracted data on study design using a standardized data extraction form. A data extraction form was developed to facilitate the electronic comparison of entry. The following data were extracted: study characteristics (including author, year of publication, percentage of patient follow-up, country, research foundation), patient characteristics (mean age), interventions characteristics (drug groups, intervention doses, and intervention duration).

### 2.5. Quality Assessment and Risk of Bias

The GRADE (Grading of Recommendations, Assessment Development and Evaluations) approach, supported by an electronic tool (McMaster University and Evidence Prime. https://gdt.gradepro.org (accessed on 21 February 2021)) [13], was used in the evidence quality assessment for the primary outcome-duration of diarrhea, and the secondary outcome-frequency of diarrhea on specific days. The quality of evidence (also known as certainty of the evidence) was categorized into four categories based on the likelihood of bias, the directness of evidence, and the consistency and accuracy of estimates: high, moderate, low, and extremely low.

The risk of bias was assessed using the revised Cochrane risk-of-bias tool. The risk-of-bias parameters included the type of randomization process (selection bias), allocation concealment (selection bias), blinding of participants and personnel (performance bias), blinding of outcome assessment (detection bias), and incomplete outcome data (attrition bias). Furthermore, biases, such as selective reporting (reporting bias) and other forms of prejudice, were taken into consideration. Due to a lack of available information, an item was classified as having an ambiguous risk of bias if it could not be assessed [14].

### 2.6. Statistical Analysis

The mean difference (MD) and 95% confidence interval (95% CI) were used as the metrics of choice for treatment effects with random-effects models. Pooled data was assessed using available case analysis. Data were analyzed for every participant for whom the outcome was obtained, rather than intention-to-treat analysis with imputation. The presence of heterogeneity among studies was measured with χ^2^ test, with a *p*-value of up to 0.10 considered significant. To measure consistency, we used the I^²^ test. An I^2^ value of 0% indicates that there has been no observed heterogeneity, while rising values imply growing heterogeneity in the data. I^2^ values above 50% suggest a significant degree of heterogeneity.

When at least ten randomized controlled trials were available, funnel plots were used to examine reporting biases for the primary outcome. The presence of reporting bias was investigated with Begg and Egger tests [15]; *p* < 0.05 implies publication bias. This was accomplished via the use of funnel plots and other visual estimation techniques. Statistical analyses were performed using STATA software (Version 15.1).

The data were analyzed using Review Manager (Version 5.4. the Nordic Cochrane Centre, the Cochrane Collaboration, Copenhagen, Denmark, 2020). For change in duration of diarrhea, the mean duration was used as reported. When reported, the standard deviation (SD) was utilized or computed from the median and interquartile range (IQR). In four trials reporting median and IQR [10,11,16,17], missing means and SDs were estimated using the formula recommended by Wan et al. [18] and McGrath et al. [19].

Four separate pre-specified subgroup analyses were performed for the primary outcomes according to the presence of potential trial-level effect modifiers:Probiotic species (individual probiotic strains *Lactobacillus acidophilus* vs. *Lactobacillus acidophilus* + other probiotic strains)The daily dose of *Lactobacillus acidophilus* (high dose [≥10^10^ CFU/day] vs. [10^9^~10^10^ CFU/day] vs. low dose [<10^10^ CFU/day])Setting (studies carried out in geographical areas, Asia vs. Europe vs. America)Etiology of diarrhea (rotavirus-positive 100% vs. rotavirus-positive 1~99% vs. unknown etiology).

## 3. Results

### 3.1. Search Results

For a flow diagram documenting the identification process for eligible trials, see Figure 1. Detailed characteristics of the included RCTs are presented in Appendix A, and excluded trials are shown in Appendix A. Ultimately, 15 RCTs that randomized 1765 participants (895 in the experimental group and 870 in the control group) were included [10,11,12,16,17,20,21,22,23,24,25,26,27,28,29].

### 3.2. Characteristics of Included Studies

The participants in the studies varied in age from 1 month to 12 years. The number of people that took part in the trials varied from 46 to 290. Included trials were carried out in Thailand [27], India [29], Peru [16], Iran [19,25], Vietnam [10], TaiWan [23], Israel [24], France [28], Belgium [17,26], Turkey [21], Indonesia [11], Korea [12], and China [22]. All the research studies considered were single-center trials.

The most common administered daily dose of *Lactobacillus acidophilus* ranged from 10^9^ to 10^10^ CFU [12,17,21,23,24,26]. Other daily doses were <10^9^ CFU [10,11], and ≥10^10^ CFU [16,25,27,28,29]. The doses were unknown in two trials. The duration of the intervention was inconsistent, lasting 2 days [27], 3 days [12,28,29], 5 days [10,21], 7 days [11,17,26], or for an unspecified period [16,19,22,23,24,25].

The blank contrast group were placebo in 6 trials and no *Lactobacillus acidophilus* in the remaining trials. In all studies, *Lactobacillus acidophilus* was used in addition to rehydration therapy, consisting of an oral rehydration solution and/or intravenous rehydration.

### 3.3. Assessment of Risk of Bias and Publication Bias

Most trials were at risk of bias for at least one of the domains. The appendix presents the findings of the risk-of-bias evaluation (see Appendix A). Only 8 (53%) of the trials adequately generated their randomization sequence, while just 11 (73%) adequately concealed allocation, and 12 (80%) trials blinded all involved parties (e.g., participants, study personnel and outcome assessors). A total of 14 trials (93% of all trials) provided complete outcome data, defined as at least 80% follow-up. Only 5 trials were considered at low risk for overall risk of bias.

GRADE analysis for the selected outcomes is presented in Appendix A.

High levels of statistical heterogeneity (I^2^ ≥ 50%) were found for both the primary outcome (the duration of diarrhea [I^2^ = 94%]) and the secondary outcome (the frequency of diarrhea at various times interval [I^2^ = 84%]). Publication bias was formally evaluated only in the analysis of the duration of diarrhea. There was publication bias for this outcome (Egger test’s test *p* = 0.015; see Appendix A). Because of the limited number of papers included in the analysis (<10), publication bias was not officially examined using a funnel plot for other outcomes.

### 3.4. Results of Included Studies

A summary of the results is presented in Table 1.

#### 3.4.1. Reporting of Diarrhea Duration

The primary outcomes are summarized in Figure 2. After conducting 15 trials with 1765 participants, researchers discovered that those who received *Lactobacillus acidophilus* had shorter diarrhea duration than those who were treated with placebo or received no treatment (MD −0.69 days, 95% CI −1.04 to −0.33; high heterogeneity [I^2^ = 94%]).

As intended, all the pre-planned subgroup analyses were performed.

Probiotic species

There appeared to be a decrease in the duration of diarrhea in children treated both in the individual probiotic strain *Lactobacillus acidophilus* subgroup (6 RCTs, *n* = 698, MD −0.47 days, 95% CI −0.95 to 0.01) and in the *Lactobacillus acidophilus* + other probiotic strains subgroup (9 RCTs, *n* = 1067, MD −0.91 days, 95% CI −1.23 to −0.59); however, the former effect was not statistically significance between the probiotic group and the placebo group (Figure 2). The test for subgroup differences suggested that there was no significant difference (*p* = 0.14), meaning that combined other probiotic strains may not modify the treatment effect. However, a smaller group of trials and participants contributed to the *Lactobacillus acidophilus* subgroup than to the *Lactobacillus acidophilus* + other probiotic strains subgroup. There was also heterogeneity between the trials within each subgroup. Overall, whether there is no efficacy of *Lactobacillus acidophilus*, or whether there is an actual significant subgroup effect, requires further investigation. The subdivided probiotic mixture was analyzed by subgroup (Appendix A).

Dose

*Lactobacillus acidophilus* was effective when used at a daily dose ≥ 10^10^ CFU (5 RCTs, *n* = 408, MD −0.69 days, 95% CI −1.25 to −0.14) and 10^9^~10^10^ CFU (6 RCTs, *n* = 588, MD −0.89 days, 95% CI −1.36 to −0.42); the latter dosage produced results of more significance (Figure 3). However, there was no reduction nor increment in the duration of diarrhea when used at a daily dose <10^9^ CFU (2 RCTs, *n* = 402, MD 0.55 days, 95% CI 0.20 to 0.91). The test for subgroup differences suggests a significant difference (*p* < 0.00001), meaning that the daily dose of probiotic may modify the treatment effect. Nevertheless, a smaller group of trials and participants contributed to the <10^9^ CFU subgroup (2 RCTs, *n* = 402) than to the ≥10^9^ CFU subgroup (11 RCTs, *n* = 988), indicating that the analysis may expand the differences between subgroups. The same result occurred in the individual probiotic strain *Lactobacillus acidophilus* (Appendix A).

Setting

There appeared to be a reduction in the duration of diarrhea in children, treated both in Asia (11 RCTs, *n* = 1451, MD −0.66 days, 95% CI −1.07 to −0.26) and in Europe (3 RCTs, *n* = 237, MD −0.88 days, 95% CI −1.41 to −0.36), but *Lactobacillus acidophilus* had no effect on the duration of diarrhea in America (1 RCT, *n* = 77, MD −0.65 days, 95% CI −1.33 to 0.03) (Appendix A). The test for subgroup differences suggested no significant difference (*p* = 0.78), meaning that geographic setting may not modify the treatment effect. However, an RCT, and a smaller group of participants, contributed to the America subgroup compared to the non-America subgroup. Overall, to establish whether there is an actual significant subgroup effect requires more trials.

Etiology

Concerning etiology, limited data indicate that *Lactobacillus acidophilus* was effective in treating diarrhea due to rotavirus-positive 100% (1 RCT, *n* = 57, MD −1.23 days, 95% CI −1.88 to −0.58), rotavirus-positive 1%~99% (6 RCTs, *n* = 718, MD −0.79 days, 95% CI −1.43 to −0.15) and unknown causes (8 RCTs, *n* = 990, MD −0.91 days, 95% CI −1.24 to −0.58) (Appendix A). The test for subgroup differences suggested no significant difference between groups (*p* = 0.60). However, each subgroup had a very small number of trials and individuals, making it difficult to draw clear conclusions.

Sensitivity Analyses

Pre-planned subgroup analyses based on trial methodological quality were carried out. Statistically significant between the studies, heterogeneity persisted in subgroup analyses, suggesting that the outcomes were not sensitive to the studies’ methodological quality, and that the outcomes were not sensitive to the classification according to the quality of methodology (Appendix A).

#### 3.4.2. Diarrhea Frequency

The included trials often reported diarrhea frequency. The studies showed a reduction in the frequency of diarrhea for those treated with *Lactobacillus acidophilus* compared with controls at all time intervals; however, it was just statistically significant at day 3 (4 RCTs, *n* = 594, MD −0.61 days, 95% CI −1.00 to −0.23) (Appendix A).

## 4. Discussion

This analysis was based on 15 trials, including 1765 children with acute gastroenteritis randomly assigned to *Lactobacillus acidophilus* or placebo. The present meta-analysis of *Lactobacillus acidophilus* for treating acute gastroenteritis in children substantially expanded on previous meta-analyses [30].

The addition of *Lactobacillus acidophilus* to standard rehydration therapy compared with placebo or no treatment was associated with a reduced duration of diarrhea by approximately 17 h (0.69 days). A subset of patients that was more likely to benefit included subjects treated with an effective daily dose of *Lactobacillus acidophilus* of ≥10^9^ CFU/day. *Lactobacillus acidophilus* had the same effect when used in European countries and Asian countries. Limited evidence suggests that *Lactobacillus acidophilus* was more effective in treating diarrhea of rotavirus origin. Only a small number of studies looked at the impact of *Lactobacillus acidophilus* supplementation on the frequency of diarrheal episodes in children. While diarrhea frequency was decreased in general, it was statistically and clinically significant only on day 3, indicating that the optimum benefit of *Lactobacillus acidophilus* in lowering diarrhea frequency may be attained at 72 h after the intervention begins.

However, caution is needed when interpreting these results, as two included studies [20,22] did not report the dose of each probiotic strain in the intervention; besides this dose issue, the course of acute gastroenteritis could also affect the efficacy of the probiotic. We think this is the most comprehensive analysis of this probiotic strain that has been conducted to date, and we hope that our findings will be especially useful to decision-makers. Because of inaccuracy, which was a key problem influencing confidence, and because there was considerable unexplained variability across studies, many component-effect estimates might be reinforced by further data, despite the large amount of data analyzed.

The present study would benefit from clarification of the uncertainties, for example, around the supplementation of zinc. Clinical studies have consistently observed an association between zinc deficiency and acute diarrhea in early childhood [31]. For babies and children suffering from severe diarrhea in resource-limited areas, the World Health Organization (WHO) advises that they receive zinc supplements [32]. Zinc supplementation was also observed to shorten the average duration of acute diarrhea in children in a meta-analysis [33]. However, in a network meta-analysis of randomized and quasi-randomized trials comparing interventions for acute diarrhea and gastroenteritis, zinc was not effective in reducing the duration of diarrhea in children in high-income countries [9]. There is still controversy around the actual efficacy of zinc for the management of acute diarrhea. In our study, zinc was added to three RCTs [11,22,24] and was absent in the other RCTs. Therefore, it remains to be established if additional zinc provision will benefit acute diarrhea and will have any influence on analysis results. The included trials covered most children. Children from both developing and developed countries aged from 1 month to 12 years old participated. However, there were some issues with the under-reporting of trial details in this area. We were missing data from many of the studies included, with data on specific population characteristics (e.g., outcomes in different ages, feeding pattern-breastfeeding, or complementary feeding) particularly poorly reported. Trials need to be subdivided in terms of their goals and objectives. The microbiome is very different between individuals; this is true not only for viral species but also for tribes and individual strains. For example, a study which applied metagenomic analysis to fecal samples from infants and their mothers showed that the samples clustered according to age, and demonstrated that the 12-month-old infant samples were most similar to the mothers. There was a more complex and less heterogeneous community as a function of time. The rising complexity was also supported by increased numbers of microbial genomes identified in the older infants [34]. It would be of benefit to stratify study populations further if large groups are included, such as stratifying children from birth to 4 months, 4 to 12 months, and over 12 months, to allow for better analysis as well as to determine whether the gastrointestinal tracts of children are distinctive at the initiation of trials. Research has shown that human milk contains, not only macro- and micronutrients, but also living cells, growth factors, and immunoprotective substances [35,36]. The use of human milk lowers the risk of gastrointestinal illness compared with complimentary food or milk powder, and it may be of particular importance when evaluating the impact of interventions on groups with different diets. Due to a lack of reporting regarding population characteristics, we were unable to determine with any certainty whether population characteristics influenced the comparative effectiveness of the interventions evaluated. Individual participant data (particularly for population characteristics) meta-analysis, or network meta-analysis using original datasets, may be the best way to investigate this in the future.

Moreover, we believe that the effects of probiotic use will vary across viral causes, but there is insufficient etiological data to support this. In our study, the rotavirus infection status reported for only seven trials [10,12,16,23,27,28,29], of which only one trial [12] tested all participants for rotavirus infection. Furthermore, owing to missing data on rotavirus vaccination status in the included trials, it is unclear whether rotavirus vaccine moderates probiotic effectiveness.

A probiotic strain is considered to have specificity. Thus, a probiotic product used in clinical trials should record the genus, species, and strain designation clearly; all trials included performed well in this aspect.

The included trials also need to collect more outcome data, such as duration of vomiting, daycare absenteeism, the rate of household transmission and cost-effectiveness, as these may affect final decisions on curative probiotic supplementation for children.

### 4.1. Quality of the Evidence

Given that the number of RCTs and participants was not impressive, the quality of evidence supporting key findings was moderate to low. The majority of our component effect estimates were downgraded by one or two levels due to imprecision, owing to the different definitions of acute gastroenteritis/diarrhea, the different operational standards, and the different definitions of the reported outcomes among trials. For example, the different time of the initiation of probiotic administration (duration of diarrhea less than 48 h, or 72 h) or the different definition of diarrhea cure (stool met the Bristol criteria or not) caused imprecision and inconsistency as defined by GRADE.

More than a half of the trials adequately generated their randomization sequence (e.g., sealed envelopes, computer-generated randomization lists, random permuted blocks); almost 73% of the trials adequately concealed allocation; 80% of the trials blinded all the parties involved, e.g., participants, study personnel and outcome assessors; and 93% provided complete outcome data. Eight included trials were considered to be at low risk of bias. In terms of certainty of evidence on outcomes, we have moderate confidence in effect estimates for the duration of diarrhea based on GRADE assessments.

### 4.2. Strengths and Limitations

The findings of this study, taken as a whole, contradict the findings of several of the low-risk bias studies that were included. There were a variety of factors that contributed to these variances. For example, the number of participants in Badriul 2015 [11] was 112, Myeong 2017 [12] was 57, and Vikrant 2005 [29] was 98; all the participant groups were not large enough, thus, the findings, whether positive or negative, might be (non)significant by chance only.

In 2019, a meta-analysis by Hania et al. [37] concluded that *Lactobacillus rhamnosus GG* can reduce the duration of diarrhea when used at a daily dose ≥10^10^ CFU (11 RCTs, *n* = 2764, MD −0.83 days, 95% CI −1.17 to −0.49) or <10^10^ CFU (4 RCTs, *n* = 1056, MD −0.92 days, 95% CI −1.83 to −0.02). This finding contradicted the results of Badriul 2015 [11], whose intervention included the probiotic strains *Lactobacillus acidophilus* 0.1 × 10^9^ CFU/day and *Lactobacillus rhamnosus GG* 1.9 × 10^9^ CFU/day. Further studies are needed to assess the effect of probiotic mixtures of *Lactobacillus acidophilus* and *Lactobacillus rhamnosus GG*. The lack of an effect of *Lactobacillus acidophilus* in Badriul 2015 [11] and Tran 2018 [10] studies may also be explained by the fact that, in contrast to other studies, a low daily dose of the probiotic (<10^9^ CFU) was administered. *Lactobacillus acidophilus* was not beneficial in treating children with acute watery diarrhea, probably because the minimum effective concentration of *Lactobacillus acidophilus* was not reached.

In 2014, guidelines for the management of acute gastroenteritis in children published by the European Society for Paediatric Gastroenterology, Hepatology and Nutrition [3] concluded that ‘the use of the probiotic *Lactobacillus rhamnosus GG* and *Saccharomyces boulardii* may be considered in the management of children with acute gastroenteritis in addition to rehydration therapy.’ Two large, well-conducted meta-analyses showed that *Lactobacillus rhamnosus GG* and *Saccharomyces boulardii* can be effective in treating children with acute gastroenteritis [37,38]. However, the European Medicines Agency (EMA) has issued a recent warning about a potential risk of fungaemia caused by *Saccharomyces boulardii* in seriously ill or immunocompromised patients, and it has recommended that *Saccharomyces boulardii* be used with caution due to the possibility of airborne contamination. In conclusion, because of safety concerns associated with the use of *Saccharomyces boulardii*, we recommend the probiotic strains *Lactobacillus acidophilus* and *Lactobacillus rhamnosus GG* to children with acute gastroenteritis, whether the probiotic strains are used in combination or alone.

While the methodology of this systematic review was robust, the conclusions are mostly constrained by the number of papers that were accessible. There are some limitations to this review. The first is the possibility of bias in some of the trials that have been included.

Second, unexplainable heterogeneity between individual trials is a significant limitation. Clinical variations and methodological issues are among the factors contributing to the heterogeneity. On clinical differences, some subgroup analyses were conducted to investigate whether factors, such as dose, setting or etiology, modified the effect of the treatment. The operational standards for each trial were not uniform, which caused other kinds of clinical differences, such as inconsistency in zinc supplementation, different definitions of diarrhea, different durations of symptoms before the intervention, and different definitions of the reported outcomes. Such diversity and inconsistency in the outcomes, combined with the lack of standardized definitions, pose a challenge in meta-analyses and should be taken into account when interpreting the findings. On methodological issues, certain subgroup analyses were carried out in order to determine if the quality of the methodology affected the treatment impact. Overall, while some of the analyses revealed a statistically significant subgroup effect and high heterogeneity between findings within each group, the small number of trials and participants contributing to each subgroup resulted in uncertainty about whether these subgroup differences matter.

Due to the differences in clinical characteristics, or the fact that some outcomes were evaluated in only a subset of trials with a limited number of participants, meta-regression is underpowered to detect small associations or diversities; meta-regression results should hence be interpreted with caution. For example, in our study, the finding that etiology had no influence on treatment effects may have been due to the limited power with which we were able to identify them.

## 5. Conclusions

The epidemiology literature is becoming saturated with probiotic and gastrointestinal disease studies, and the use of probiotic strains is being developed more meticulously. In summary, our findings suggest *Lactobacillus acidophilus* treatment of acute gastroenteritis in children is effective when administered at a daily dosage of ≥10^9^ CFU/day. But there are differences in the efficacy and acceptability profiles across *Lactobacillus acidophilus*, *Lactobacillus rhamnosus* and *Saccharomyces boulardii*, especially safety issues with the use of *Saccharomyces boulardii*. We recommend the probiotic strains *Lactobacillus acidophilus* and *Lactobacillus rhamnosus* for children with acute gastroenteritis, whether the probiotic strains are used in combination or alone. We hope that this analysis contributes a helpful perspective to aid in these decisions. Whether probiotic recommendations for acute gastroenteritis in children should be tailored based on susceptibility factors (for example, age, feeding pattern, whether vaccinated against rotavirus) remains unanswered and will be important to advance knowledge in the field.

## Figures and Tables

**Figure 1 nutrients-14-00682-f001:**
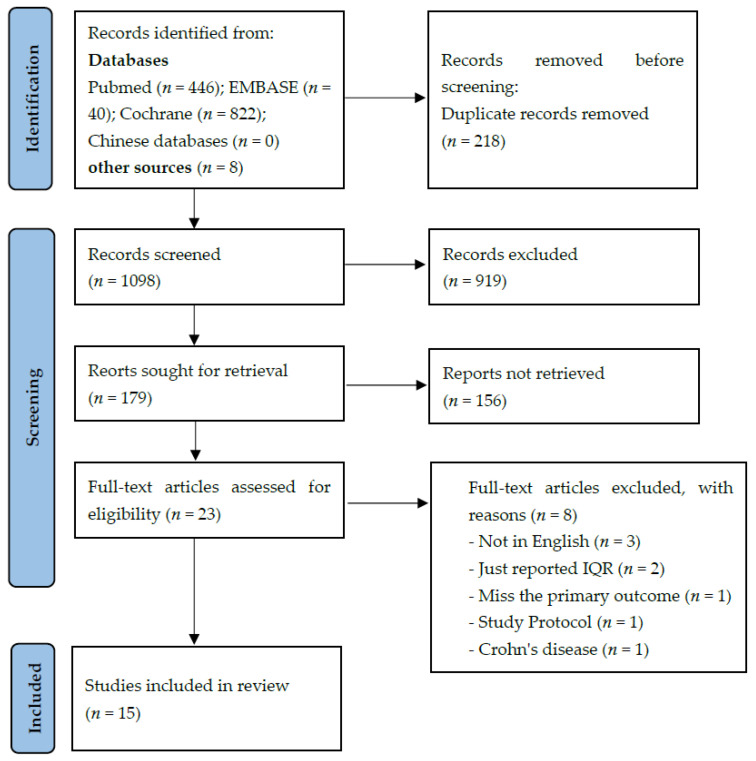
Flow diagram of search strategy and study selection.

**Figure 2 nutrients-14-00682-f002:**
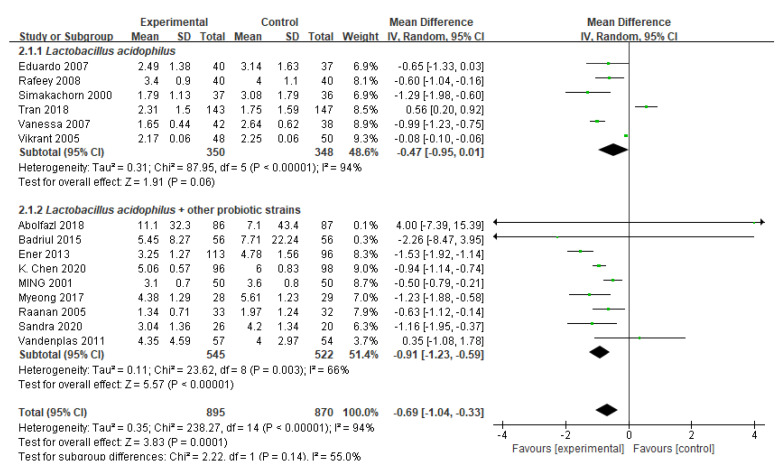
Forest plot of randomized controlled trials of *Lactobacillus acidophilus* vs. control in acute gastroenteritis. Effect on duration of diarrhea (days). Probiotic species.

**Figure 3 nutrients-14-00682-f003:**
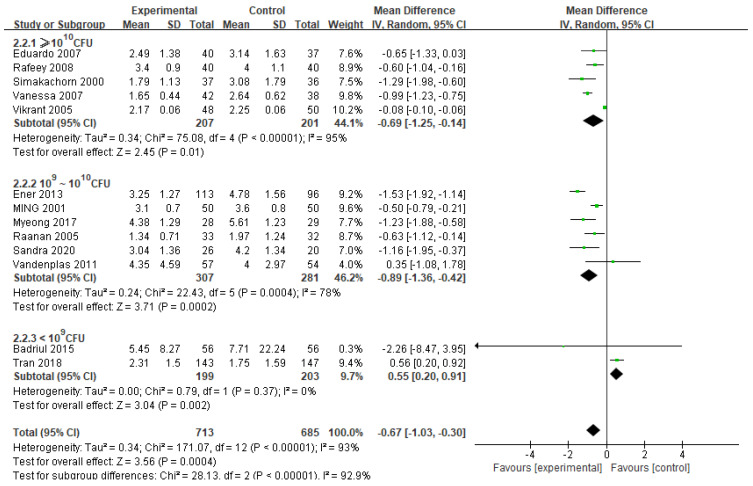
Forest plot of randomized controlled trials of *Lactobacillus acidophilus* vs. control in acute gastroenteritis. Effect on duration of diarrhea (days). Dose.

**Table 1 nutrients-14-00682-t001:** Overview of the results.

Outcome or Subgroup	RCT (*n*)	Participants (*n*)	Statistical Method, Random Effect Model	Effect Estimate (95% CI)	Heterogeneity, I^2^ (%)
Duration of diarrhea (days)	15	1765	MD	−0.69 [−1.04, −0.33]	94%
Probiotic strains					
individual probiotic strains *Lactobacillus acidophilus*	6	698	MD	−0.47 [−0.95, 0.01]	94%
*Lactobacillus acidophilus* + other probiotic strains	9	1067	MD	−0.91 [−1.23, −0.59]	66%
Daily dose of *Lactobacillus acidophilus* in all studies					
≥10^10^ CFU	5	408	MD	−0.69 [−1.25, −0.14]	95%
10^9^~10^10^ CFU	6	588	MD	−0.89 [−1.36, −0.42]	78%
<10^9^ CFU	2	402	MD	0.55[0.20, 0.91]	0%
Daily dose of *Lactobacillus acidophilus* in individual probiotic strains studies					
≥10^9^ CFU	5	408	MD	−0.69 [−1.25, −0.14]	95%
<10^9^ CFU	1	290	MD	0.56[0.20, 0.92]	N/A
Setting					
Asia	11	1451	MD	−0.66 [−1.07, −0.26]	94%
Europe	3	237	MD	−0.88 [−1.41, −0.36]	43%
America	1	77	MD	−0.65 [−1.33, 0.03]	N/A
Etiology					
Rotavirus-positive 100%	1	57	MD	−1.23 [−1.88, −0.58]	N/A
Rotavirus-positive 1%~99%	6	718	MD	−0.79 [−1.43, −0.15]	94%
Unknown etiology	8	990	MD	−0.91 [−1.24, −0.58]	59%
Frequency of diarrhea					
On day 1	6	694	MD	−0.47 [−1.71, 0.78]	92%
On day 2	7	965	MD	−0.45 [−1.22, 0.33]	79%
On day 3	4	594	MD	−0.61 [−1.00, −0.23]	64%
On day 4	3	493	MD	−0.23 [−0.61, 0.16]	0%
On day 5	1	209	MD	0.15 [−0.22, 0.52]	N/A

Abbreviations: RCT, randomized controlled trial; 95% CI, 95% confidence interval; MD, mean difference; N/A, not applicable.

## Data Availability

Data sharing not applicable.

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
