# Peer review of "A Systematic Review and Meta-Analysis: Lactobacillus acidophilus for Treating Acute Gastroenteritis in Children"

_nutrients, 2022, doi:10.3390/nu14030682_

Round 1
Reviewer 1 Report
It is helpful to decide which probiotics use for acute gastroenteritis in children. I believe this is first time to research as RCT with Lactobacillus acidophilus for acute gastroenteritis in children.
1, Please spell check by native.
2, Please add explanation about the characteristics about Lactobacillus acidophilus more in detail such ph is acid which means easily reach to small intestine, thrive in small intestine rather than colon and metabolism influence to epithelial cells etc.
3, Microbiome is so difference in individual. That is not only for viral spices but also tribe and individual. I recommend to add those comment.
Author Response
Dear reviewer,
Thank you for your comments concerning our manuscript entitled “A systematic review and meta-analysis Lactobacillus acidophilus for treating acute gastroenteritis in children” (Manuscript ID 1536305). Those comments are all valuable and very helpful for revising and improving our paper. We have studied the comments carefully and have made corrections according to the suggestions. Revised portions are marked in red in the manuscript. The main corrections and the response to the reviewer's comments are as follows. We appreciate for Reviewers' warm work earnestly, and hope that the correction will meet with approval.
Sincerely Yours
Libo Zhao
Department of Pharmacy Peking University Third Hospital
Tel/Fax: 13366972232
E-mail address: libozhao2011@163.com (L. Zhao)
Response to the editor:
1, Please spell check by native.
Response: Thank you for your reminder. According to your advice, this manuscript has been edited by an English independent suppliers of editing services: WeiFeng educational agency.
2, Please add explanation about the characteristics about Lactobacillus acidophilus more in detail such ph is acid which means easily reach to small intestine, thrive in small intestine rather than colon and metabolism influence to epithelial cells etc.
Response: Thank you for your suggestions. We have supplemented some explanation about the characteristics about Lactobacillus acidophilus, and we have supplemented some mechanisms of action of lactobacillus acidophilus.
Line:47-53
Lactobacillus acidophilus achieved maximum in survival rates and CFU/ml after adaptation at pH 4.2 and 5.0 [4], accordingly, it can easily reach to small intestine (pH<7), thrive in small intestine rather than colon (pH>7). Possible mechanisms for the therapeutic effects of lactobacillus acidophilus include amelioration of the impairment of electrolyte absorption [5], maintain immunological homeostasis [6], antibiotic activity/antiviral activity [7] and promotion of the intestinal epithelial integrity [8].
References:
[4] Lorca G.L.; Raya R.R.; Taranto M.P.; Valdez de G.F. Adaptive acid tolerance response in Lactobacillus acidophilus. Biotechnology Letters. 1998, 20(3), 239-241. [CrossRef]
[5] Borthakur A.; Gill R.K.; Tyagi S.; Koutsouris A.; Alrefai W.A.; Hecht G.A.; Ramaswamy K.; Dudeja P.K. The probiotic Lactobacillus acidophilus stimulates chloride/hydroxyl exchange activity in human intestinal epithelial cells. J Nutr. 2008, 138(7), 1355-9. [CrossRef]
[6] Azevedo M.S.; Zhang W.; Wen K.; Gonzalez A.M.; Saif L.J.; Yousef A.E.; Yuan L. Lactobacillus acidophilus and Lactobacillus reuteri modulate cytokine responses in gnotobiotic pigs infected with human rotavirus. Benef Microbes. 2012, 3(1), 33-42. [CrossRef]
[7] Liévin-Le Moal V. A gastrointestinal anti-infectious biotherapeutic agent: the heat-treated Lactobacillus LB. Therap Adv Gastroenterol. 2016. 9(1), 57-75. [CrossRef]
[8] Seth A.; Yan F.; Polk D.B.; Rao R.K. Probiotics ameliorate the hydrogen peroxide-induced epithelial barrier disruption by a PKC- and MAP kinase-dependent mechanism. Am J Physiol Gastrointest Liver Physiol. 2008, 294(4), G1060-9. [PubMed]
3, Microbiome is so difference in individual. That is not only for viral spices but also tribe and individual. I recommend to add those comment.
Response: Thank you for your suggestions. We have supplemented some comment on “Microbiome is so difference in individual”, and we have corrected the age group “birth to 4, 4 to 12 months, and over 12 months”.
Line:358-364
Microbiome is so difference in individual. That is not only for viral spices but also tribe and individual. For example: A study applied metagenomic analysis on fecal samples from infants and their mothers showed that the samples clustered according to age, and demonstrated that the 12-month-old infant samples were most similar to the mothers. There was a more complex and less heterogeneous community as a function of time. The rising complexity was also supported by increased numbers of microbial genomes identified in the older infants [34].
References:
[34] Bäckhed F.; Roswall J.; Peng Y.; Feng Q.; Jia H.; Kovatcheva-Datchary P.; Li Y.; Xia Y.; Xie H.; Zhong H.; et al. Dynamics and Stabilization of the Human Gut Microbiome during the First Year of Life. Cell Host Microbe. 2015, 10, 17(6), 852. [PubMed]

Reviewer 2 Report
The introduction of the article should also include considerations regarding the mechanism of action of lactobacillus acidophilus in acute gastroenteritis with diarrhea in children. The authors want the practitioners to consider using probiotics with lactobacillus acidophilus as a therapeutic solution, so considerations are needed regarding the mechanisms of action on intestinal microbiota We understand that the studies included in the meta-analysis are very heterogeneous, but the results obtained with other probiotic species should be presented in comparison. We appreciate the work methodology and the statistical analysis part. .
Author Response
Dear reviewer,
Thank you for your comments concerning our manuscript entitled “A systematic review and meta-analysis Lactobacillus acidophilus for treating acute gastroenteritis in children” (Manuscript ID 1536305). Those comments are all valuable and very helpful for revising and improving our paper. We have studied the comments carefully and have made corrections according to the suggestions. Revised portions are marked in red in the manuscript. The main corrections and the response to the reviewer's comments are as follows. We appreciate for Reviewers' warm work earnestly, and hope that the correction will meet with approval.
Sincerely Yours
Libo Zhao
Department of Pharmacy Peking University Third Hospital
Tel/Fax: 13366972232
E-mail address: libozhao2011@163.com (L. Zhao)
Response to the editor:
- The introduction of the article should also include considerations regarding the mechanism of action of lactobacillus acidophilus in acute gastroenteritis with diarrhea in children. The authors want the practitioners to consider using probiotics with lactobacillus acidophilus as a therapeutic solution, so considerations are needed regarding the mechanisms of action on intestinal microbiota.
Response: Thank you for your suggestions. We have supplemented the mechanism of action of lactobacillus acidophilus in acute gastroenteritis with diarrhea in children.
Line:49-53
Possible mechanisms for the therapeutic effects of lactobacillus acidophilus include amelioration of the impairment of electrolyte absorption [5], maintain immunological homeo-stasis [6], antibiotic activity/antiviral activity [7] and promotion of the intestinal epithelial integrity [8].
References:
[5] Borthakur A.; Gill R.K.; Tyagi S.; Koutsouris A.; Alrefai W.A.; Hecht G.A.; Ramaswamy K.; Dudeja P.K. The probiotic Lactobacillus acidophilus stimulates chloride/hydroxyl exchange activity in human intestinal epithelial cells. J Nutr. 2008, 138(7), 1355-9. [CrossRef]
[6] Azevedo M.S.; Zhang W.; Wen K.; Gonzalez A.M.; Saif L.J.; Yousef A.E.; Yuan L. Lactobacillus acidophilus and Lactobacillus reuteri modulate cytokine responses in gnotobiotic pigs infected with human rotavirus. Benef Microbes. 2012, 3(1), 33-42. [CrossRef]
[7] Liévin-Le Moal V. A gastrointestinal anti-infectious biotherapeutic agent: the heat-treated Lactobacillus LB. Therap Adv Gastroenterol. 2016. 9(1), 57-75. [CrossRef]
[8] Seth A.; Yan F.; Polk D.B.; Rao R.K. Probiotics ameliorate the hydrogen peroxide-induced epithelial barrier disruption by a PKC- and MAP kinase-dependent mechanism. Am J Physiol Gastrointest Liver Physiol. 2008, 294(4), G1060-9. [PubMed]
- We understand that the studies included in the meta-analysis are very heterogeneous, but the results obtained with other probiotic species should be presented in comparison. We appreciate the work methodology and the statistical analysis part.
Response: Thank you for your advices and the affirmation to our work. We have made a comparison between probiotic species in the SUPPORTING INFORMATION, for full details, see Figure S3. The experimental group was Lactobacillus acidophilus, Lactobacillus acidophilus + Bifidobacterium, Lactobacillus acidophilus + Lactobacillus rhamnosus, and Lactobacillus acidophilus + other probiotic strains respectively; the control group were placebo or no Lactobacillus acidophilus (oral rehydration solution or no intervention) in the remaining trials., but there are still heterogeneous in these subgroup analyses.

Round 2
Reviewer 2 Report
No additional comments. I think the paper is ready to be published. I sincerely appreciate the author's initiative to do this meta-analysis regarding the role of lactobacillus acidophilus in the treatment of acute gastroenteritis in children. Even though the efficacy of this probiotic take action after 48 -72h is still very important for the practioners to know that it could be usefull in shortener the diareea disease.